# Aberrant Expression of Androgen Receptor Associated with High Cancer Risk and Extrathyroidal Extension in Papillary Thyroid Carcinoma

**DOI:** 10.3390/cancers12051109

**Published:** 2020-04-29

**Authors:** Chen-Kai Chou, Shun-Yu Chi, Fong-Fu Chou, Shun-Chen Huang, Jia-He Wang, Chueh-Chen Chen, Hong-Yo Kang

**Affiliations:** 1Division of Endocrinology and Metabolism, Department of Internal Medicine, Kaohsiung Chang Gung Memorial Hospital, Chang Gung University College of Medicine, Kaohsiung City 83301, Taiwan; 2Department of Surgery, Kaohsiung Chang Gung Memorial Hospital, Chang Gung University College of Medicine, Kaohsiung City 83301, Taiwan; 3Department of Pathology, Kaohsiung Chang Gung Memorial Hospital, Chang Gung University College of Medicine, Kaohsiung City 83301, Taiwan; 4Graduate Institute of Clinical Medical Sciences, Chang Gung University, Kaohsiung City 83301, Taiwan; 5Department of Obstetrics and Gynecology, Kaohsiung Chang Gung Memorial Hospital, Chang Gung University College of Medicine, Kaohsiung City 83301, Taiwan

**Keywords:** papillary thyroid carcinoma, androgen receptor, tumorigenesis, sex difference, epithelial-mesenchymal transition

## Abstract

Male gender is a risk factor for mortality in patients with papillary thyroid carcinoma (PTC). This study investigated the impact of androgen receptor (*AR*) gene expression on the clinical features and progression of PTC. The levels of *AR* mRNA and protein in frozen, formalin-fixed, paraffin-embedded tissue samples from PTC and adjacent normal thyroid tissue were assessed by quantitative real-time polymerase chain reaction and immunohistochemical staining, respectively, and the relationships between *AR* expression and clinical features were analyzed. The thyroid cancer cell lines, BCPAP and TPC-1, were used to evaluate the effects of *AR* on the regulation of cell migration, and key epithelial–mesenchymal transition (EMT) markers. *AR* mRNA expression was significantly higher in normal thyroid tissue from men than women. The sex difference in *AR* mRNA expression diminished during PTC tumorigenesis, as *AR* mRNA expression levels were lower in PTC than normal thyroid tissues from both men and women. *AR* mRNA expression was significantly decreased in PTC patients with higher risk and in those with extrathyroidal extension. Overexpression of *AR* in BCPAP cells decreased cell migration and repressed the EMT process by down-regulating mRNA expression of *N-cadherin*, *Snail1*, *Snail2*, *Vimentin*, and *TWIST1* and up-regulating *E-cadherin* gene expression. These results suggest that suppression of the androgen–*AR* axis may lead to aggressive tumor behavior in patients with PTC.

## 1. Introduction

Thyroid cancer is the most common endocrine malignancy, with increasing prevalence over time [1,2,3]. Papillary thyroid carcinoma (PTC) accounts for around 80% of all thyroid cancers [4]. The incidence of PTC varies by age and sex; women have a higher age-adjusted incidence rate with a ratio of women to men of 3–5.5:1 [2,5]. PTC generally have a good prognosis [3], although aggressive risk factors, including old age, male sex, huge tumor size, and extra-thyroidal and distant metastases, are associated with poorer prognosis [6,7]. Since male gender is a risk factor for poor clinical outcome and mortality in patients with PTC [8], the reproductive hormones may link to PTC’s clinical behavior and cause [5]. However, few studies have addressed the molecular mechanisms account for sex hormone signaling pathways in PTC.

The sex hormone receptors are a group of steroid hormone receptors that interact with sex hormones, including androgens, estrogen, and progesterone [9]. Sex steroids exert their functions by binding to their cognate nuclear steroid receptors, a class of transcription factors that regulate the expression of specific genes, including those involving critical biological functions in men and women. While the important role of sex hormones in cancer gender disparity is well documented and easily explained for sex-specific cancers such as breast and prostate cancers [10,11], the sex differences by which sex steroid receptors contribute to the gender disparity on clinical cause and outcome of PTC remain a promising but not fully understood research area. Although estrogen and estrogen receptors (α and β) are involved in the pathology of thyroid cancer [12,13] and androgen receptor (*AR*) has been first evidenced for the expression of both normal thyroid tissue and malignant thyroid tumors [14,15,16], less is known about the roles of these sex hormone receptors in the development and progression of PTC. 

These intriguing observations suggest that sex steroids and/or their receptors may play central roles in regulating the development of thyroid cancers. Better understanding of the molecular mechanisms of sex hormone receptors involved in PTC initiation and progression may aid in the identification of novel therapeutic targets and molecular markers. The current study presents the evidence to show that low expression of *AR* is associated with the development and progression of PTC and *AR* has tumor suppressive function in the metastasis of PTC.

## 2. Results

### 2.1. Lower Level of AR Expression is Present in PTC than in Normal Thyroid Tissue

An initial cohort with paired cancer and normal tissue specimens were prospectively collected from 71 patients with PTC, all of whom received standard treatment, including surgery, radioactive iodine therapy, and thyroid hormone therapy, at the Chang Gung Memorial Hospital-Kaohsiung Medical Center. The levels of expression of mRNAs encoding various sex hormone receptors, including *AR*, estrogen receptor alpha *(ERα)*, estrogen receptor beta (*ERβ*), and progesterone receptor *(PR)*, were analyzed in these samples by qRT-PCR. While there is marginal or no significant difference in the *ERα*, *ERβ,* and *PR* mRNA expression between PTC and the surrounding normal parenchyma, the level of *AR* mRNA expression was significantly lower in PTC than in the surrounding normal parenchyma (*p* < 0.001; Figure 1A). In majority of PTC samples (60/71 = 84.5%), *AR* mRNA expression was significantly lower in cancerous areas than in normal areas (Figure 1B). Microarray-based RNA profiling analysis using the Gene Expression Omnibus (GEO) datasets were further validated that the expression levels of *AR* in the PTC were relatively low compared to other normal human thyroid tissues (Appendix A).

### 2.2. Decreased Expression of AR is Associated with Advanced Clinical Characteristics of PTC 

Understanding the molecular mechanisms that regulate the biology of PTC in men and women may help identify novel targets for therapeutic intervention. Although *AR* was shown to play an important role in prostate and breast cancers [11,17], its differential role in men and women with PTC remains unclear. To test whether *AR* mRNA expression differs in PTC from men and women, *AR* mRNA expression was further analyzed in an expanded cohort with 137 patients with PTC, including 108 women and 29 men (Table 1), and the correlations between *AR* mRNA expression levels and the demographic and clinical characteristics of these patients were assessed. Of these 137 patients, 66 (48.2%) had extrathyroidal extensions, 59 (43.0%) had lymph node metastases, and 30 (21.9%) were classified as a high risk group. 

*AR* mRNA expression was higher in normal thyroid glands of men than women (Figure 2A). In contrast, *AR* mRNA expression did not differ significantly in PTC specimens of men and women. *AR* mRNA expression levels were significantly lower in most PTC specimens from both men and women group than in paired normal thyroid tissues (*p* < 0.05; Figure 2A). To test whether *AR* mRNA expression was associated with a more aggressive thyroid tumor phenotype, correlations between *AR* mRNA expression and clinical features in this cohort of patients with PTC were analyzed. *AR* mRNA expression was significantly lower in tumors from high risk than low risk patients, as well as being significantly lower in patients with than without extrathyroidal extension (*p* < 0.05 each; Table 2, Figure 2B). The pattern of *AR* mRNA expression did not differ significantly among different age groups or between patients with and without lymph node metastasis (Figure 2B). We also performed logistic regression model to analyze the role of *AR* in extrathyroidal extension. The statistical analysis (odds ratio and 95% confidence interval, respectively) showed that there were 3 independent risk factors for extrathyroidal extension, including age (>55 y/o) (OR: 2.88, 95% CI: 1.15–7.2, *p* = 0.024); lymph node metastasis (OR: 8.56, 95% CI: 3.69–19.94, *p* < 0.001); and expression of *AR* (OR: 0.38, 95% CI: 0.17-0.89, *p* = 0.026), suggesting that *AR* expression is an independent factor for extrathyroidal extension in PTC (Table 3). While serum concentration of total and free testosterone was higher in male patients than female patients and the serum free testosterone level in male patients with PTC (6.66 ± 2.39 ng/dL, mean age: 51.35 y/o) is obviously below than normal male population (Reference range: 6.76–22.76 ng/dL in male), it was not significantly associated with clinical characteristics (data in preparation).

AR protein expression was validated by IHC in 38 randomly selected pairs of normal and thyroid cancer tissue (Figure 3A), using the scoring classification described in Appendix A. AR protein expression was significantly lower in thyroid cancer tissue than in paired normal tissue, both overall and in men and women separately (*p* < 0.05 each; Figure 3B,C). Similar to *AR* mRNA expression patterns, AR protein expression did not show the sex difference when compared the PTC specimens from men and women (Figure 3C). Analysis of the associations between AR protein expression and clinical features in patients with PTC showed that a trend of reduced AR protein expression tended to be associated with advanced tumor characteristics, including higher tumor stage, extrathyroidal extension and lymph node metastasis (Appendix A). 

### 2.3. AR Decreases the Cancer Cell Migratory Activity of PTCs In Vitro

To investigate whether *AR* expression is involved in the behavior of PTC, mRNA levels of *AR* expression were assessed in various thyroid cell lines. We found that *AR* mRNA expression was significantly lower in the thyroid cancer cell lines BCPAP and TPC-1 than in the normal human thyroid cell line Nthy-ori-3-1 (Figure 4A). To assess the role of *AR* in PTC cells, we transfected a vector overexpressing *AR* (pSG5-AR) or a control vector (pSG5) into BCPAP and TPC-1 cells. Transfection of pSG5-AR, but not pSG5, markedly enhanced the expression of *AR* mRNA and protein (Figure 4B and Appendix A). Analysis of changes in function of thyroid tumor cells, using an in-vitro gain of *AR* function model, showed that overexpression of *AR* in BCPAP and TPC-1 cells reduced cell migration (Figure 4C,D). 

### 2.4. AR Reduces the Epithelial–Mesenchymal Transition (EMT) Process in PTC 

Because *AR* expression was significantly lower in PTC with than without extrathyroidal extension and *AR* over-expression significantly reduced tumor cell migration, we hypothesized that the tumor suppressive function of *AR* in PTC may be involved in inhibition of EMT, a process characterized by the transition of epithelial cells to mesenchymal stem cells, with increased migratory and invasive activities. To test this hypothesis, we analyzed the mRNA expression of several important EMT markers, including *E-cadherin*, *N-cadherin*, *Snail1*, *Snail2*, *TWIST1,* and vimentin, in PTC cell lines overexpressing *AR*. We found that the mRNA levels of expression of *N-cadherin*, *Snail-1*, *Snail-2*, *vimentin,* and *TWIST1* were significantly lower, and the mRNA level of *E-cadherin* significantly higher, in BCPAP cells overexpressing *AR* than in control cells (*p* < 0.05 each; Figure 5). Testing of the association of *E-cadherin* mRNA expression with clinical risk factors in original 71 PTC samples showed that advanced cancer stage and extrathyroidal extension were associated with significantly lower *E-cadherin* expression (Appendix A), providing further evidence that *AR* suppresses EMT during PTC tumorigenesis. To illustrate the association of E-cadherin and *AR* expression on PTC, we analyzed the mRNA expression of *E-cadherin* and *AR* expression in these 71 prospectively collected PTC samples and paired normal tissues by RT-PCR analysis. As shown in Appendix A, the scatter diagram confirmed that *AR* and *E-cadherin* expression levels were positively correlated (*p* < 0.05).

## 3. Discussion

The effects of sex hormones are mediated by hormone-specific nuclear receptors that regulate gene expression, tumor cell biology [18] and sex-specific pathology [19]. Although *AR* has crucial roles in many types of cancer, as shown by sex difference [17,20], the influence of *AR* was less understood in PTC. The present study found that *AR* mRNA expression in paired normal thyroid tissue was higher in men than women but it was significantly lower in PTC specimens in both men and women. We further found that reduced *AR* mRNA expression was associated with high cancer risk and extrathyroidal extension in patients in PTC (Table 2). Moreover, overexpression of *AR* inhibits the PTC cell migration activity and decreases the EMT markers (Figure 4 and Figure 5); thereby suggesting that decreasing *AR* expression may promote the EMT process during the initiation of tumor metastasis or progression. These results not only indicate that the reduction of *AR* levels has specific link to the increasing risk of developing thyroid cancer in the neoplastic process, but also play a critical role in regulating tumor behavior associated with extrathyroidal extension in PTC. 

The molecular mechanisms of loss of *AR* actions contributing to advanced tumor behavior remain unclear. PTC overexpressing *AR* cells showed higher expression of *E-cadherin* and lower expression of *N-cadherin*, *Snail-1&2*, *TWIST1,* and *vimentin* than cells not overexpressing *AR* (Figure 5). The dysregulation of EMT associated markers has been demonstrated to correlate with clinical outcome of PTC [21,22]. An IHC report also demonstrated loss of E-cadherin correlated with tumor progression, such as: Distant metastasis, local recurrence and advanced tumor stage [23]. The activated *AR* had been proved to bind on promotor region of *E-cadherin* and cooperates with histone deacetylase 1 (*HDAC1*) which lead to regulation of EMT process [24]. In a human thyroid cancer cell lines study [25], loss of *E-cadherin* expression was shown to correlate with hypermethylation of *E-cadherin* 5’ CpG island and a similar study reported that activation of EMT by androgen deprivation therapy in metastatic prostate cancer [26]. Together with these results, we also found that advanced cancer stage and extrathyroidal extension in PTC were associated with significantly lower *AR* (Figure 2) and *E-cadherin* expression (Appendix A), providing further evidence to support that insufficiency of the *AR* signaling may promote EMT by which PTC tumor cells can adapt to promote disease recurrence and progression. *AR* expression has been shown to highly correlate with prostate cancer progression, with *AR* mutations causing hormone-refractory disease [20]. *AR*, however, was recently reported to have both oncogenic and tumor suppressive roles dependent on the differential expression pattern in different tumor cell types [27,28]. The potent androgen 5 alpha-dihydrotestosterone (5α-DHT), has been shown to inhibit the proliferation of an *AR*-positive PTC cell line in vitro [29]. In agreement with previous study [16], our preliminary data also showed that the serum free testosterone level in male patients with PTC (6.66 ± 2.39 ng/dL, mean age: 51.35 y/o) is obviously below than normal male population (Reference range: 6.76–22.76 ng/dL in male). Together, these data suggest that androgens/*AR* axis may play the potential tumor suppressive role in PTC.

*AR* had been demonstrated to reduce cancer stemness by repression of *CD44* and *SOX2* [30], on the contrary, loss of *AR* expression could upregulate *STAT3* expression and lead to promote development of cancer stemness in prostate cancer cells [31]. The cancer stemness is highly associated with advanced clinical presentation and poor prognosis in thyroid carcinoma, EMT has been shown to induce cancer stem-like cell generation and tumor progression in human thyroid cancer cells including decreased expression of *E-cadherin* [32]. High-throughput genomic and proteomic approaches to the study of the interaction between androgen–*AR* axis and the EMT pathway with cancer stemness may reveal the novel molecular mechanism underlying PTC tumorigenesis and provide useful information to find new therapeutic targets in thyroid cancers. 

The biological characteristics, pathological features and clinical behavior in PTC may differ between men and women [5,33]. A large scale of epidemiological study showed that male sex had an independent adverse effect on PTC-specific survival [34], whereas other studies found that male sex did not have an independent effect [35,36]. Our study further demonstrated that *AR* mRNA and protein levels were higher in normal thyroid tissue of men than of women, but these sex differences of *AR* expression diminished during PTC tumorigenesis (Figure 2A and Figure 3C). The similar expression pattern of *AR* in males verses females PTC in our study could not provide further insight to explain why the rates of extrathyroidal extension, lymph node metastasis, higher cancer stage and distant metastasis were found to be higher in men than in women with PTC. According to Wang et al. [8], male sex has been shown to be a robust independent risk factor for PTC-specific mortality in *BRAF^V600E^* patients but not in wild-type *BRAF* patients. It is possible that the different pathological features and clinical outcomes between women and men may be mediated by the interaction between the oncogenic pathways and the sex hormone regulation. Further studies regarding to deficiency of *AR* axis connection to PTC’s oncogenic pathway, such as *Ras/Raf/MEK/ERK* signal transduction may help to strength the molecular basis of sex disparity on PTC-specific mortality in male patients. 

Previous studies showed that *AR* activity in non-medullary thyroid carcinomas, as determined by ligand binding activity, was found to be higher in a majority of men and lower in a majority of women compared with normal thyroid tissue [16]. Furthermore, data from immunohistochemical staining showed that the expression of AR protein did not change in the majority of PTC in male compared to controls but the majority of PTC in female expressed decreased levels of AR protein [16]. Nevertheless, the present study found that *AR* expression was significantly higher in normal thyroid tissue from men than from women but these gender differences on *AR* expression diminished in PTC (Figure 2 and Figure 3) and the association between *AR* mRNA expression and PTC was further validated by analyzing a publicly available GEO microarray dataset, suggesting that discrepancies could be due to protocols for *AR* activity detection, variations in methodology, enrolled thyroid specimen type and cohort size. First, heterogenicity of immunohistochemical staining in various tumor regions may make results less accurate than quantitative RT-PCR analysis, there is evidence suggesting RNA expression patter profiles may differ to the protein expression pattern profiles and steroid ligand binding assay may have detect bias with false-positive results [37]. Second, as the rabbit polyclonal primary AR antibody (C-19): SC-815 used in that study [16] has been discontinued and replaced by monoclonal antibody AR (441): sc-7305 by Santa Cruz. The rabbit monoclonal AR antibody (clone SP107) used in this study (IHC part) is more specific and intended for in vitro diagnostic use for human, provides a stronger signal and eliminates lot-to-lot variation characteristic of most, if not all, polyclonal antibodies. While it has been reported that AR protein-positive tumors were more aggressive with capsular invasion than AR protein-negative tumors [38], the limitation of this study is lack of high risk of PTC patients with only T1 stage specimen and limited numbers of AR staining on the relative smaller sample size of cohort, which makes the role of AR in PTC’s survival and recurrence difficult to be interpreted. The evaluation of association with long-term clinical outcome and androgens/AR status is of interest to elucidate the value in prognosis implication AR on PTC.

There have been few studies reported evaluating gender discrepancies between *AR* and PTC’s outcome and prognosis. The limitation of this study is the small sample size of male subjects (*n* = 29) and short duration of follow up period (average less than 5 yrs), therefore, the role of *AR* in sex differences for PTC’s survival and recurrence is difficult to determine. The evaluation of the relationship between long-term clinical outcome and androgen–*AR* axis is on the way to further investigate.

## 4. Materials and Methods 

### 4.1. Tumor Samples and Patient Information

This prospective study analyzed 137 patients with PTC who were treated at the Kaohsiung Chang Gung Memorial Hospital, Kaohsiung, Taiwan, between August 2013 and October 2018. Samples from these patients were selected based on the clinical features of the patients and access to prospectively collected tissue collections by one of the authors (Dr. Shun-Yu Chi). The 137 patients included 108 women and 29 men, ranging in age from 18–78 years. Patients aged <18 years or >80 years were excluded, as were patients with follicular carcinoma, medullary cancer and anaplastic cancer. Details of the clinical features of the PTC patients in this study are presented in Table 1. This study was approved by the Institutional Review Board of Chang Gung Memorial Hospital (ethic code: 201601932B0). Informed consent was obtained from all included patients. Details concerning clinical data collection and tumor node-metastasis classifications of these samples have been reported previously [39]. 

Tissue samples were snap-frozen in liquid nitrogen at the time of total thyroidectomy, and subsequently stored in liquid nitrogen. The American Joint Commission on Cancer-International Union Against Cancer criteria were utilized for risk classification based on each patient’s clinicopathologic risk factors [40]. Patients aged <55 years with stage I or II PTC were defined as the low-risk group, with all remaining patients defined as the high-risk group. 

### 4.2. RNA Extraction and Reverse Transcription Polymerase Chain Reaction (RT-PCR)

Total RNA was extracted from surgical specimens which stored in DNA/RNA Shiled^TM^ (ZYMO, Irvine, CA, USA) and thyroid cancer cell lines using Quick-RNATM Mini Prep kit (ZYMO, Irvine, CA, USA). The samples were processed by the manufacturer’s instructions. The primers for real-time PCR were: *h18S*, forward (5′-GTAACCCGTTGAACCCCATT-3′) and reverse (5′-CCATCCAATCGGTAGTAGTG-3′); *AR*, forward (5′-CCTGGCTTCCGCAACTTACAC-3′) and reverse (5′-GGACTTGTGCATGCGGTACTCA-3′); *ERα*, forward (5′-AGTTGGCCGACAAGGAGTTG-3′) and reverse (5′-CGCACTTGGTCGAACAGG-3′); *ERβ*, forward (5′-TACTGACCAACCTGGCAGACAG-3′) and reverse (5′-TGGACCTGATCATGGAGGGT-3′); and *PR*, forward (5′-AGCCGGTCCGGGTGCAAG-3′) and reverse (5′-CCACCCAGAGCCCGAGGG-3′). The protocol for real-time PCR has been described [39].

### 4.3. Tissue Specimens and Immunohistochemical Staining

Surgical specimens of PTC were routinely fixed in 10% formalin, embedded in paraffin, and sectioned at a thickness of 2.5 μm. The tissue sections were deparaffinized in xylene, dehydrated in a graded ethanol series, and heated in boiling 0.01 M citrate buffer (pH 6.0) for 15 min in a microwave oven for antigen retrieval. After cooling to room temperature, endogenous peroxidase activity was blocked by incubation with 3% hydrogen peroxide for 30 min. The sections were incubated with primary rabbit monoclonal anti-AR antibody (clone SP107, Zeta Corp., Sierra Madre, CA, USA) diluted in Antibody Diluent Reagent Solution (Invitrogen, Carlsbad, CA, USA) at 4 °C overnight, rinsed in 0.1% TBS-T, stained with Dako REAL™ EnVision™ Detection System, Peroxidase/DAB, Rabbit/Mouse as a chromogen, and counterstained with hematoxylin. The sections were photographed on a Nikon Eclipse 50i microscope (Nikon, Tokyo, Japan) connected to a Nikon DS-Fi1 color digital camera (Nikon, Tokyo, Japan). Staining intensity was measured by a pathologist (Dr. Shun-Chen Huang) at Chang Gung Memorial Hospital using light microscopy. The section containing both normal thyroid and tumor part was chosen for reading. Ten low power (10X objective lens) fields of each part were selected to evaluate its intensity and percentage. The staining intensity was graded as 0, negative; 1+, weakly positive; 2+, moderately positive; 3+, strongly positive. The percentage of positively stained cells were classified as grade 1, <10%; grade 2, ≥10% and ≤25%; grade 3, >25% and ≤50%; grade 4, >50% and ≤75%; and grade 5, >75% [41]. The final IHC score was obtained by multiplying the intensity and percentage grade. Then the average staining scores of the tumor part and the normal part were compared.

### 4.4. Thyroid Cancer Cell Culture

The human thyroid cancer cell lines BCPAP (German Collection of Microorganisms and Cell Culture, Braunschweig, Germany) and TPC-1 and the normal thyroid cell line Nthy-ori-3-1 (European Collection of Authenticated Cell Cultures, Porton Down, England) were routinely cultured in RPMI1640 containing 10% fetal bovine serum (FBS), 1x antibiotic-Antimycotic (Gibico cat# 15240112), and 2 mM L-glutamine (Gibco, Carlsbad, CA, USA) at 37 °C in a humidified chamber containing 5% CO_2_. 

### 4.5. Plasmid Transfection 

The *AR* over-expression plasmid (pSG5-AR) and its control vector (pSG5) were used to transfect the BCPAP and TPC-1 cells as previously described [42,43]. Briefly, cells were seeded at a density of 3 × 10^5^ cells per well in 6-well plates in antibiotic-free medium for 24 h prior to transfection. The cells were transfected with pSG5-AR or pSG5 using DharmaFECT^TM^ transfection reagent (GE HealthCare Dharmacon, Lafayette, CO, USA). The cells were collected 24 h later, and the protein and mRNA expression of *AR* was quantified.

### 4.6. Western Blot Analysis

Cells transfected with plasmids for 24 h were lysed in RIPA buffer containing protease, phosphate inhibitor cocktail (Sigma, St. Louis, MO, USA) and 1 mM phenylmethylsulphonyl fluoride (PMSF). Samples of 100 μg total protein were separated on 8% SDS-polyacrylamide gels and transferred onto nitrocellulose membranes (GE HealthCare, Buckinghamshire, UK), which were incubated with primary mouse monoclonal anti-AR antibody (clone AR 441 Invitrogen, Carlsbad, CA, USA; 1:200) and anti-β-actin antibody (Millipore, Billerica, MA, USA, 1:10,000) at 4 °C overnight, followed by incubation with goat-anti-mouse HRP-conjugated secondary antibody at room temperature for 1 h (Cell Signaling Technology, London, UK). Bands were visualized using an Enhanced ChemoLuminescence kit (GE HealthCare, Buckinghamshire, UK) according to the manufacturer’s instructions. Relative levels of AR expression were determined by normalization to the expression of β-actin. Densitometric analysis of the protein bands was performed using Bio-Rad Quantity One 1-D Analysis software.

### 4.7. Migration Assay

Cell migration assay were performed in transwells in 24 well plates. Transwells (Falcon, New York, NY, USA) with 8uM Polyethylene Terephthalate membrane used for cell migration. Thyroid cancer cells transfected with a vector overexpressing *AR* (pSG5-AR) or a control vector (pSG5) for 24 h were seeded at 5 × 10^4^ cells/well in the upper chambers, with the lower chambers filled with culture medium containing 10% FBS, followed by incubation at 37 °C for 24 h in a humidified incubator containing 5% CO_2_. Cells that did not pass through the filter were wiped off. Cells that had migrated to the reverse side of the filter were fixed with methanol for 10 min, washed with PBS, stained with Giemsa solution (Millipore) for 30 min. Cells were photographed and counted under a light microscope with a 10x objective lens with the selective use of a 10× objective lens for confirmation. The number of migrated cells was counted by analyzing 3 random fields of the membranes per condition.

### 4.8. Statistical Analyses

Continuous variables, including *AR* expression in paired cancer and normal tissue from patients with PTC, were compared using Wilcoxon test. Correlations between *AR* expression and tumor characteristics were assessed by Mann–Whiney U-test analysis. Variables were enrolled in multiple logistic regressions with forward stepwise procedure to identify independent risk factors for extrathyroidal extension. The Spearman correlation coefficient was used to determine the correlation of *AR* and *E-cadherin* expression level. All statistical analyses were performed using SPSS software (version 19.0; SPSS, Chicago, IL, USA). All *p*-values were two sided, with *p*-values < 0.05 considered statistically significant.

## 5. Conclusions

To sum up, our data indicate that *AR* may play a pivotal role in thyroid cancer tumorigenesis, as *AR* mRNA expression levels were lower in PTC than normal thyroid tissues. In addition, the pattern of low *AR* expression was associated with high cancer risk and extrathyroidal extension during the tumor progression in papillary thyroid carcinoma. As growing number of clinical studies continue and further study is needed to further elucidate the relationship between *AR* and PTC, our data in this study suggest that the potential importance of activation of androgen–*AR* axis and its downstream signaling pathways may serve as a novel therapeutic target in advanced and metastatic thyroid carcinomas.

## Figures and Tables

**Figure 1 cancers-12-01109-f001:**
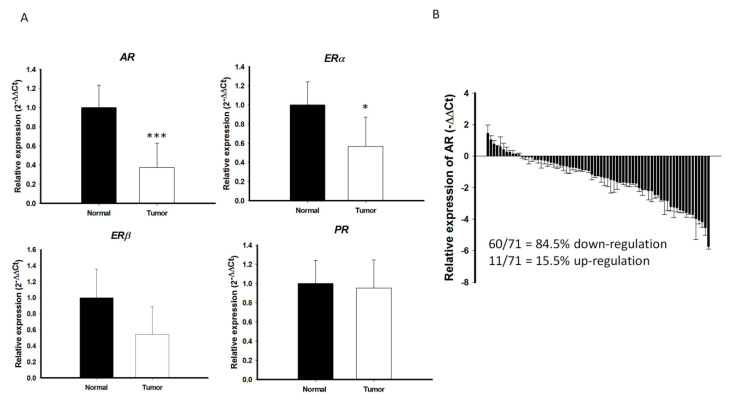
Quantitative reverse transcription–polymerase chain reaction (qRT-PCR) analysis of (**A**) *AR*, *ERα*, *ERβ,* and *PR* in PTCs (*n* = 71) and paired normal tissue. The fold change values indicate the relative change in the expression levels between samples and its internal control (*18S*), assuming that the value of *18S* expression level of each sample was equal to 1. (**B**) Relative expression of *AR* mRNA levels in PTC normalized to adjacent non-tumor tissues (T/N fold change). *AR* mRNA expression indicated a significant reduction compared to matched normal tissue. Fold change in *AR* mRNA expression was calculated relative to paired normal thyroid tissue. *** *p* < 0.001 * *p* < 0.05.

**Figure 2 cancers-12-01109-f002:**
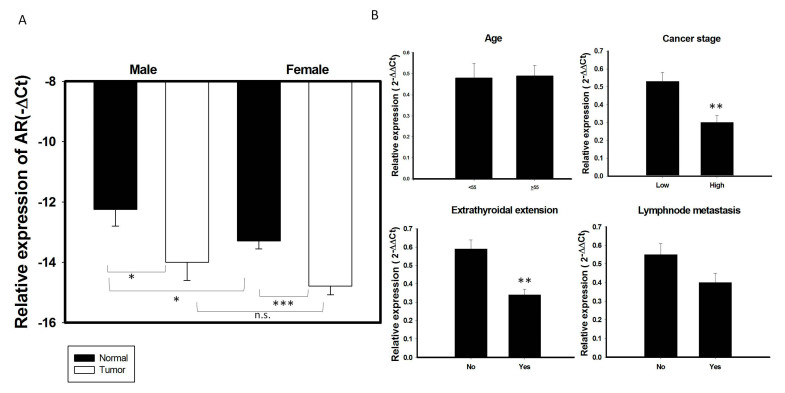
Association of *AR* mRNA expression with clinical characteristics in 137 PTC samples. (**A**) The expression of *AR* mRNA is significantly decreased in PTC tumor specimens than its adjacent normal tissue in both male and female gender patients. (**B**) Association of *AR* with clinical features in 137 PTC patients stratified as follows: age; cancer risk, low risk (stage I + II) vs. High (stage III + IV); the presence of lymph node metastasis and extrathyroidal extension.* *p* < 0.05 ** *p* < 0.01 *** *p* < 0.01 n.s: non-significant.

**Figure 3 cancers-12-01109-f003:**
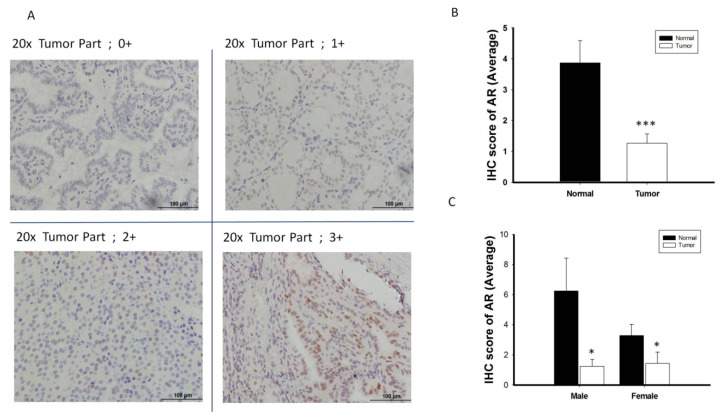
The analysis of androgen receptor (AR) protein expression level by immunohistochemistry (IHC) (*n* = 38). (**A**) Scoring classification of IHC; 0, negative (−); 1+, weakly positive; 2+, moderately positive (++); 3+, strongly positive (+++). (**B**) Normal thyroid tissue express high expression level of AR protein scoring than its paired cancerous specimen. (**C**) The expression of AR protein is significantly decreased in PTC tumor specimens than its adjacent normal tissue in both male and female gender patient. * *p* < 0.05 *** *p* < 0.001

**Figure 4 cancers-12-01109-f004:**
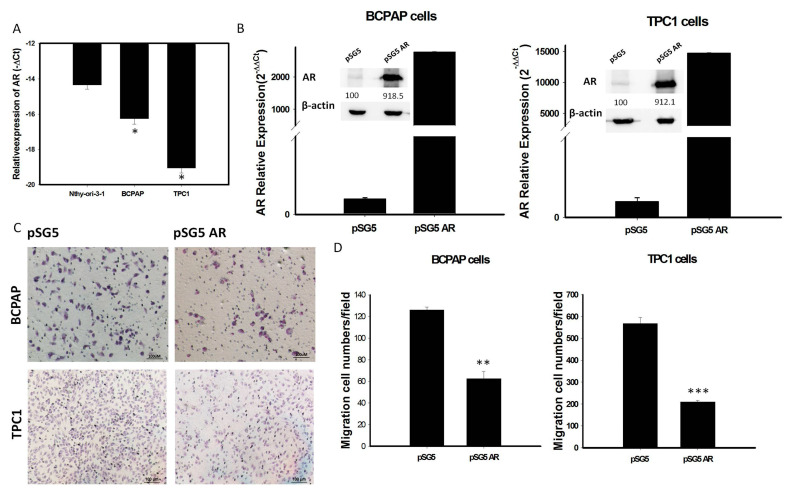
(**A**): The endogenous mRNA expression of *AR* in normal thyroid cell line (Nthy-ori-3-1) is higher than PTC-derived cancer cell lines (BCPAP and TPC1); (**B**) Quantification of mRNA and protein expression levels of AR after transfection with pSG5-AR and pSG5 in BCPAP cells and TPC-1 detected by qRT-PCR and Western blotting. The relative expression level of *AR* mRNA is presented as 2^−ΔΔCt (AR-18S)^. (**C**) Reprensentative image of the cell migratory abilities of BCPAP cells and TPC-1 transfected with pSG5-AR and pSG5 in the Transwell (Corning) migration assay. (**D**) Quantitative analysis of the migratory abilities of BCPAP and TPC-1 cells transfected with pSG5-AR and pSG5. Data are presented as the means of three independent experiments, and bars represent the SDs. * *p* < 0.05, ** *p* < 0.01, *** *p* < 0.01

**Figure 5 cancers-12-01109-f005:**
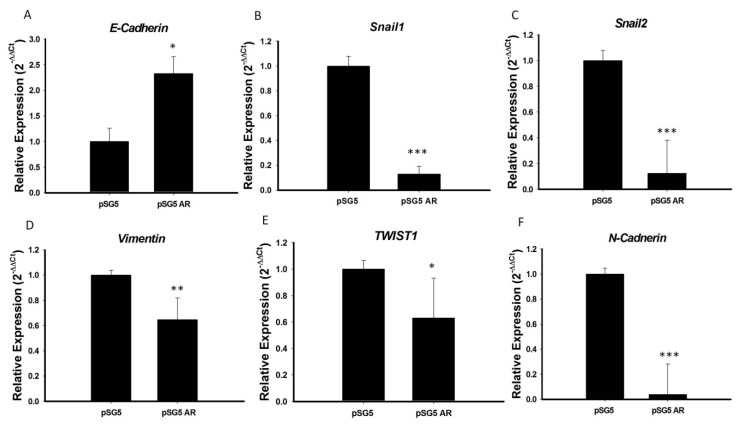
*AR* repressed EMT process by regulation of the expression of EMT marker genes. (**A**) *E-cadherin*, (**B**) *Snail1*, (**C**) *Snail2*, (**D**) *Vimentin*, (**E**) *TWIST1,* and (**F**) *N-cadherin* mRNA levels were analyzed by qRT-PCR, and *18S* mRNA was used as a control to ensure equal loading. Three independent experiments for measuring were analyzed by Western blotting. * *p* < 0.05, ** *p* < 0.01 and *** *p* < 0.001 compared with the control group.

**Table 1 cancers-12-01109-t001:** Clinicopathological features of papillary thyroid carcinomas in this study (*n* = 137).

Clinical Features	Number
Age at the time of diagnosis (years)	48.50 ± 13.37
Sex (male/female)	29/108
Tumor size (cm)	1.95 ± 1.15
Lymph node metastasis	59 (43.0%)
Extrathyroidal extension ^a^	66 (48.2%)
Gross	12 (8.7%)
Minimal	54 (39.4%)
Tumor staging (AJCC) ^b^	
Low risk	107 (78.1%)
High risk	30 (21.9%)
Distant metastasis (%)	2 (1.4%)
Tumor subtype	Classical 89
	Follicular variant 30
	Microcarcinoma 17
	Diffuse sclerosing 1

The data are shown as mean +/− SD unless otherwise indicated. a. Minimal extrathyroidal extension is defined as tumor cells extending to the sternothyroid muscle or perithyroidal soft tissue and gross extrathyroidal extension was defined as tumors extending beyond the thyroid capsule and with invasion of the subcutaneous soft tissue, larynx, trachea, esophagus, or recurrent laryngeal nerve. b. The low-risk group was defined as those patients who were less than 55 years old and had stage I papillary thyroid carcinoma (PTC) and those patients who aged 55 years or more with stage I or II PTC according to the American Joint Committee on Cancer staging system 8^th^ edition. The remaining patients were defined as high-risk group.

**Table 2 cancers-12-01109-t002:** Association of *AR* mRNA expression with clinical features in papillary thyroid carcinomas. (*n* = 137).

Clinical Features		Androgen Receptor mRNA	
		2^−(ΔΔCt)^	P
Age at the time of diagnosis (years)	<55	0.48 ± 0.07	0.35
	≥55	0.49 ± 0.05	
Sex	Male	0.50 ± 0.05	0.38
	Female	0.42 ± 0.08	
Tumor size (cm)	<3	0.50 ± 0.05	0.59
	≥3	0.48 ± 0.08	
Lymph node metastasis	No	0.55 ± 0.06	0.059
	Yes	0.40 ± 0.05	
Extrathyroidal extension	Absent	0.54 ± 0.06	0.021
	Present	0.34 ± 0.03	
Tumor staging (AJCC) ^a^	Low risk	0.53 ± 0.05	0.033
	High risk	0.30 ± 0.04	
Tumor subtype	Classical	0.44 ± 0.05	n.s ^a^
	Follicular variant	0.54 ± 0.04	0.63
	Microcarcinoma	0.62 ± 0.05	

The data are shown as mean +/− SE. ^a^ n.s: non-significant between different tumor subtypes.

**Table 3 cancers-12-01109-t003:** Multiple logistic regression analysis of factors associated with extrathyroidal extension.

Variable	OR (95% CI)	*p*
Age at diagnosis ^a^	2.88 (1.15–7.2)	0.024
Sex ^b^	0.52 (0.18–1.47)	0.22
Tumor size ^c^	0.86 (0.31–2.32)	0.762
Lymph node metastasis	8.56 (3.69–19.94)	<0.001
*AR* expression ^d^	0.38 (0.17–0.89)	0.026

a. ≥55 y/o vs. <55 y/o. b. Male vs. female. c. ≥3 cm vs. <3 cm. d. defined by expression level below/above the median level; high vs. low.

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
