# Peer review of "Aberrant Expression of Androgen Receptor Associated with High Cancer Risk and Extrathyroidal Extension in Papillary Thyroid Carcinoma"

_cancers, 2020, doi:10.3390/cancers12051109_

Round 1

Reviewer 1 Report

The present reported study is in global experimentally well-conducted.

Some precisions are required concerning the cell culture conditions. Did the authors add androgens in cell culture media?

If yes, which molecule has been used and what was the final concentration, and the exposure duration? As the two cell lines have been transfected to overexpress AR, it is important to show that the observed consequences are AR-dependent. Also, different levels of AR activation could reinforce the data presented.

The quality of submitted figures and legends is relatively poor and should be clearly improved.

Author Response

  1. Some precisions are required concerning the cell culture conditions. Did the authors add androgens in cell culture media?

If yes, which molecule has been used and what was the final concentration, and the exposure duration? As the two cell lines have been transfected to overexpress AR, it is important to show that the observed consequences are AR-dependent. Also, different levels of AR activation could reinforce the data presented.

Response 1: Thanks for your comment. We did not add androgen (DHT or testosterone) in cell culture medium in our experiments. Indeed, we totally agree with the reviewer that it is important to show that the observed consequences are AR-dependent. Since the endogenous expression level of AR in these PTC cell lines is very low, it is difficult to knockdown the AR expression to demonstrate that the observed consequences are AR-dependent. Therefore, we overexpressed AR in PTC cell lines to observe the changes of cell migration and the expression levels of major EMT genes in the way indicative of AR-mediated consequences.

  1. The quality of submitted figures and legends is relatively poor and should be clearly improved.

Reply 2: Thanks for your kind suggestion. We had re-made our figures and improved the quality of these figures in this new version to make it much clear than previous one.

Reviewer 2 Report

The work investigates AR expression in PTC and assesses its clinicopathological correlations. The AR mRNA level was found to be higher in normal thyroid tissue of male patients with PTC than in females. However, AR expression was diminished in most tumor tissues, in which no statistically significant difference in expression between males and females was detected. Similar observations were made in FFPE tissues using IHC for the AR protein. AR mRNA expression was associated with extrathyroidal extension and higher clinical stage of PTC, although this could not be statistically confirmed on IHC despite the apparent trends. Transient overexpression of AR in PTC cell lines decreased cell migration and changed the expression levels of major EMT genes in the way indicative of EMT weakening. Relevant molecular mechanisms of AR-induced changes in EMT gene expression remained beyond the scope of the study being mentioned in the Discussion part.

It was concluded that low AR expression may help in identifying high-risk PTC patients, and the authors proposed that targeting the androgen-AR axis may provide therapeutic advantage.

While the study could not provide clues as to why male PTC patients have higher risk of mortality than females, the findings are of potential (yet vague) clinical relevance.

Several points need to be addressed.

  1. Please use the official gene names and italicize those when the gene (i.e., not a protein) is meant throughout the text.
  2. Please analyze additional GEO data sets (e.g. GSE35570, GSE50901) and NGS data sets (e.g. TCGA) to provide more evidence of AR downregulation in PTC and the difference in the expression level in thyroids in males and females.
  3. What were histological subtypes (PTC variants) of tumors in the study? Were there correlations with the AR expression?
  4. Please make clear whether “Extrathyroidal extension” includes both minimal and gross one. If such information is available, please compare AR expression level in tumors with minimal or gross extension.
  5. Was there a correlation between the relative mRNA level and IHC score/intensity/grade? These data can be added as supplementary material.
  6. P. 7 of 16/L10: “2.3 AR decreases the cancer cell behaviors of PTCs in vitro” is confusing, please rephrase.
  7. What is the reason for using different reference housekeeping genes: 18S (p. 3 of 16/L26 and Methods p. 12 of 16/L14), GAPDH (p. 8 of 16/L6) and b-actin (p. 9 of 16/L5)? Please clarify.
  8. Methods, IHC (p. 12 of 16). Please add the methodology of grading and staining intensity evaluation: how many fields and microscope magnification.
  9. Methods, cell culture (p. 13 of 16/L1). Was streptomycin added to the medium?
  10. Methods, plasmid (p.13 of 16). What was the source of pSG5-AR? Please add brief description of cloning procedure if the vector was prepared in-house or where it was received/purchased from.
  11. Methods, Western blotting (p.13 of 16). “Relative levels of AR expression were determined by normalization to the expression of β-actin.” Were band intensities measured by densitometry/ image analysis? What software was used?
  12. Methods, migration assay (p.13 of 16). Please add the culture plate/well type here (as in p. 8 of 16) and how many fields and microscope magnification were used to count cells.
  13. Methods, statistical analysis (p.13 of 16). Paired or independent sample t-tests were used to assess clinical correlations of AR expression level. Did the distributions of the expression levels conform with approximate normality? How this was examined – please add this information. If there were obvious departures from normality, corresponding non-parametric tests should be used.
  14. Methods, statistical analysis (p.13 of 16). To obtain more solid evidence of AR expression association with clinical parameters it is strongly recommended to use multivariate regression analysis in order to account for other variables potentially affecting the correlation.
  15. Conclusion (p.13 of 16). This short part needs to be rigorously revised.

- The “…our data indicate that AR may play a protective role in normal thyroid cells” statement is out of context as this was not addressed in the study; only the drop in AR expression in PTC tissue was demonstrated as the authors correctly mention at the end of this sentence, “…it is reduced in PTC”.

- “In addition, the pattern of low AR expression may be used to identify high-risk PTC patients, as well as being a tumor progression marker.” What would be the test parameters (reference range, cut-off value, PPV, NPV, etc.?, in male and female patients separately?)? No data related to PTC prognosis are presented too. Thus, the results seem to be too preliminary for this, especially for “Conclusions”.

- The “… targeting on activation of androgen-AR axis and its downstream signaling pathways may serve as a novel therapeutic strategy in PTC.” This part of the sentence is highly speculative and needs to be either modified appropriately or removed. What kind of therapeutic strategy is meant? If the long-term strategy, particularly applicably to mortality, then there is no such data in the work as the authors acknowledge (p. 11 of 16/L 30). Again, too preliminary.

Author Response

Response to Reviewer 2 Comments

  1. Please use the official gene names and italicize those when the gene (i.e., not a protein) is meant throughout the text.

Reply1: Thanks for your kind suggestion. We had checked the genes names and italicized them throughout the new version of manuscript.

  1. Please analyze additional GEO data sets (e.g. GSE35570, GSE50901) and NGS data sets (e.g. TCGA) to provide more evidence of AR downregulation in PTC and the difference in the expression level in thyroids in males and females.

Reply: Thanks for your kind suggestion. We had downloaded two microarray datasets from NCBI GEO, including GSE50901 and GSE35570.

Among the GSE35570 dataset, 45 normal and 32 tumor microarray chips were analysed. We inputted the microarray raw data into Partek software. As a result, we can derive the p-value and fold change of AR gene between specified comparisons. We noticed that the expression levels of AR in the PTC were relatively low compared to other normal human thyroid tissues is GSE 35570. Thus, we also add this new data in supplementary figure 1B according to the current analysis.

Most microarray chips of GSE50901 belong to tumor tissues, with age and gender information available. Then, with the default parameters specified, we conducted ANOVA based on male vs. female or tumor vs. normal comparison. Among the GSE50901 dataset, 35 female tumor and 14 male tumor microarray chips were analysed. We noticed that the expression levels of AR in the male PTC were relatively low compared to female PTC is GSE 35570. This new data provide the evidence to show the significant difference of the AR expression level between male and female PTCs. Due to time limit, we are on the way to fully analyse thyroid cancer data sets from TCGA.

  1. What were histological subtypes (PTC variants) of tumors in the study? Were there correlations with the AR expression?

Reply: Thanks for your kind suggestion. There are 89 classical PTC, 28 follicular variant PTC, 17 microcarcinoma and 1 diffuse sclerosing PTC in our cohort. However, there are no expression differences between these PTC histologic subtypes. We had modified our table1 and table2 to provide this information.

  1. Please make clear whether “Extrathyroidal extension” includes both minimal and gross one. If such information is available, please compare AR expression level in tumors with minimal or gross extension.

Reply: Thanks for your kind suggestion. Those with extrathyroid extensions were classified minimal extrathyroidal extension, which is defined as tumor cells extending to the sternothyroid muscle or perithyroidal soft tissue and gross extrathyroidal extension was defined as tumors extending beyond the thyroid capsule and with invasion of the subcutaneous soft tissue, larynx, trachea, esophagus, or recurrent laryngeal nerve as AJCC 8TH edition’s recommendation.

In our cohort, 8.7% and 39.4% were defined as gross and minimal extrathyroid extensions. However, we see no AR  expression (2–(ΔΔCt) ) difference between tumors with minimal (0.35 ± 0.04) or gross extension (0.31 ± 0.06) due to limited case number in gross extension group (12/137 = 8.7%).

  1. Was there a correlation between the relative mRNA level and IHC score/intensity/grade? These data can be added as supplementary material.

Reply: Thanks for your comment. We analysed the relative mRNA level and IHC score/intensity/grade according to the reviewer’s suggestion. However, there no significant association (p-value=0.4) between relative mRNA level and IHC score due to the majority (>70%) patient’s IHC score are less than 2.

  1. P. 7 of 16/L10: “2.3 AR decreases the cancer cell behaviors of PTCs in vitro” is confusing, please rephrase.

Reply: Thanks for your kind suggestion. We had modified our phrase to “AR decreases the cancer cell migratory activity of PTCs in vitro” on P8/L9.

  1. What is the reason for using different reference housekeeping genes: 18S (p. 3 of 16/L26 and Methods p. 12 of 16/L14), GAPDH (p. 8 of 16/L6) and b-actin (p. 9 of 16/L5)? Please clarify.

Reply: Thanks for your kind suggestion. We used 18S as housekeeping genes in RT-PCR experiment. We used b-actin as housekeeping genes in western blot experiments. We had corrected these mistakes in figure 4 (P8) and figure 5 (P9/L26).

  1. Methods, IHC (p. 12 of 16). Please add the methodology of grading and staining intensity evaluation: how many fields and microscope magnification.

Reply: Thanks for your kind suggestion. We add the detailed methodology description as below (P13/L13-20): The section containing both normal thyroid and tumor part was chosen for reading. Ten low power (10X objective lens) fields of each part were selected to evaluate its intensity and percentage. The staining intensity was graded as 0, negative; 1+, weakly positive ; 2+, moderately positive; 3+, strongly positive. The percentage of positively stained cells were classified as grade 1, <10%; grade 2, ≥10% and ≤25%; grade 3, >25% and ≤50%; grade 4, >50% and ≤75%; and grade 5, >75%. The final IHC score was obtained by multiplying the intensity and percentage grade. Then the average staining scores of the tumor part and the normal part were compared.

  1. Methods, cell culture (p. 13 of 16/L1). Was streptomycin added to the medium?

Reply: Thanks for your kind suggestion. We added the Antibiotic-Antimycotic, 100X (Gibico cat# 15240112) which contains streptomycin in our cell medium. We had also corrected the mistake in methodology part (page13/L27).

  1. Methods, plasmid (p.13 of 16). What was the source of pSG5-AR? Please add brief description of cloning procedure if the vector was prepared in-house or where it was received/purchased from.

Reply: Thanks for your kind suggestion. pSG5‐AR plasmid used in this study  is a human androgen receptor expression vector obtained from Dr. Chawnshang Chang in the University of Rochester, New York, USA (J Biol Chem. 1999 Mar 26;274(13):8570-6.). Molecular characterization of pSG5‐AR had been previously described in “The prostate Volume58, Issue41 March 2004 Pages 319-324. We have described and added these two references in the Methods section.

  1. Methods, Western blotting (p.13 of 16). “Relative levels of AR expression were determined by normalization to the expression of β-actin.” Were band intensities measured by densitometry/ image analysis? What software was used?

Reply: Thanks for your kind suggestion. We had measured band intensity with densitometric analysis of the protein bands was performed using Bio-Rad Quantity One 1-D Analysis software. (P14/L6-7).

  1. Methods, migration assay (p.13 of 16). Please add the culture plate/well type here (as in p. 8 of 16) and how many fields and microscope magnification were used to count cells.

Reply: Thanks for your kind suggestion. Cell migration assay were performed in transwells in 24 well plates. Transwells (Falcon) with 8uM Polyethylene Terephthalate membrane was used for cell migration. Cells were photographed and counted under a light microscope with a 10x objective lens with the selective use of a 10x objective lens for confirmation. The number of migrated cells was counted analysing 3 random fields of the membranes per condition. (P14/L10-11 and 17-19).

  1. Methods, statistical analysis (p.13 of 16). Paired or independent sample t-tests were used to assess clinical correlations of AR expression level. Did the distributions of the expression levels conform with approximate normality? How this was examined – please add this information. If there were obvious departures from normality, corresponding non-parametric tests should be used.

Reply: Thanks for your kind suggestion. The AR mRNA expression was not normal distribution (Shapiro-Wilk test and Kolmogorov-Smirnov test; p-value were both < 0.001). So we analysed the association of AR mRNA expression with clinical features in papillary thyroid carcinomas by Mann–Whitney U test. We also modified the result and methodology part on P5-6 and P14/L23-24.

  1. Methods, statistical analysis (p.13 of 16). To obtain more solid evidence of AR expression association with clinical parameters it is strongly recommended to use multivariate regression analysis in order to account for other variables potentially affecting the correlation.

Reply: Thanks for your kind suggestion. We use multivariate linear regression analysis and find that extrathyroid extension is an independent risk factor for low level of AR expression. This result had been modified on table 2 and P5/L18-20.

  1. Conclusion (p.13 of 16). This short part needs to be rigorously revised.

Reply: Thanks for your kind suggestion. We had modified our abstract and conclusion part to fully address the impact of AR dysregulation in PTC in this study. (P14/L30-37)

Round 2

Reviewer 2 Report

The authors have answered most questions, and the manuscript has been substantially improved. A few points still need corrections.

  1. The last sentence of the Abstract (p. 2 of 17, l. 19-20) says “These results suggest that inhibition of the androgen-AR axis may have the poor prognosis outcomes in patients with PTC.”. Since this work did not address “prognosis outcomes”, this term is unfounded here. Please rephrase the statement, pointing out that suppression of the androgen-AR axis associates with more aggressive tumor behavior.
  2. Please use uniform terminology, either “extrathyroid extension” (e.g. as in the article title) or “extrathyroidal extension” (e.g. as p.5 of 17, l. 2-3, Table 2, etc.) throughout the text. “Extrathyroidal extension” would be more common.
  3. Please check carefully throughout the text that the gene name “AR” is italicized when it relates to mRNA (e.g. p.5 of 17, l. 10-12).
  4. A brief description of the multivariate regression model needs to be added to the methods. Furthermore, if I understood correctly the explanation on p.5 of 17, l. 18-20, the authors used the AR expression (2–(ΔΔCt) as a continuous dependent variable, and extrathyroidal extension and tumor staging as explanatory variables which were tested in a multivariate linear regression model. If this is so, the approach is very confusing as the cause and effect are reciprocally swapped. It appears that the AR expression is attempted to be explained by extrathyroidal extension instead of explaining the latter by the AR expression.

What would be necessary to do is to create a logistic regression model in which extrathyroidal extension is an outcome (Yes/No, the dependent variable), and the AR expression is one of explanatory variables. In the model, the AR expression can be tested as a continuous variable or it can be transformed in a dummy variable using e.g. expression level below/above the median or falling into different terciles/quartiles/etc. Besides the AR expression, other variables listed in Tables 1 and 2 could be entered in a model (usually logistic regression models are adjusted at least for age and sex). Then the variable selection can be performed using any reasonable algorithm, either automatic or manual, to see whether the AR expression remains significant.

It is expected that the odds ratio estimate and its 95% confidence interval are reported for the variable of interest (if it remains in the optimal model) along with these for other variables in the model. The p-values can be reported too.

The authors are encouraged to consult with a statistician in case the regression modelling is of a difficulty.

  1. In Supplementary Materials and Methods (p. 1, l. 5), please correct “GSE3678 dataset. We analyzed..” à “GSE3678 dataset, we analyzed…”.

Author Response

Response to Reviewer 2 Comments

The authors have answered most questions, and the manuscript has been substantially improved. A few points still need corrections.

  1. The last sentence of the Abstract (p. 2 of 17, l. 19-20) says “These results suggest that inhibition of the androgen-AR axis may have the poor prognosis outcomes in patients with PTC.”. Since this work did not address “prognosis outcomes”, this term is unfounded here. Please rephrase the statement, pointing out that suppression of the androgen-AR axis associates with more aggressive tumor behavior.

Response 1: Thanks for your comment. In abstract section, we had modified our phrases to emphasize that suppression of the androgen-AR axis may lead to more aggressive tumor behavior in PTC (P2, L19-20).

  1. Please use uniform terminology, either “extrathyroid extension” (e.g. as in the article title) or “extrathyroidal extension” (e.g. as p.5 of 17, l. 2-3, Table 2, etc.) throughout the text. “Extrathyroidal extension” would be more common.

Response 2: Thanks for your comment. We had uniformed the term of “extrathyroidal extension” in our manuscript (P1,L3; P2,L13 and L19; P4,L18 and Table1; P5, L2-3 ,L17; P6 L2 and Table 2; Table3; P7, L9 and L16; P9, L14  and L24 ; P10, L18 and L23; P11,L9, L38; P15,L9 and L17).

  1. Please check carefully throughout the text that the gene name “AR” is italicized when it relates to mRNA (e.g. p.5 of 17, l. 10-12).

Response 3: Thanks for your comment. We had italicized the gene name “AR” when it relateds to mRNA (P2, L3; P3, L24; P4, L6 and L15; P5, L10-11 and L20 and Table2; P6 Table3 ; P7, L4 ; P8, L15-16 ; P9, L2 ; P10, L2 and L25; P14, L13, L18 and L37 ; P15, L5 and L15)

  1. A brief description of the multivariate regression model needs to be added to the methods. Furthermore, if I understood correctly the explanation on p.5 of 17, l. 18-20, the authors used the AR expression (2–(ΔΔCt) as a continuous dependent variable, and extrathyroidal extension and tumor staging as explanatory variables which were tested in a multivariate linear regression model. If this is so, the approach is very confusing as the cause and effect are reciprocally swapped. It appears that the AR expression is attempted to be explained by extrathyroidal extension instead of explaining the latter by the AR expression.

What would be necessary to do is to create a logistic regression model in which extrathyroidal extension is an outcome (Yes/No, the dependent variable), and the AR expression is one of explanatory variables. In the model, the AR expression can be tested as a continuous variable or it can be transformed in a dummy variable using e.g. expression level below/above the median or falling into different terciles/quartiles/etc. Besides the AR expression, other variables listed in Tables 1 and 2 could be entered in a model (usually logistic regression models are adjusted at least for age and sex). Then the variable selection can be performed using any reasonable algorithm, either automatic or manual, to see whether the AR expression remains significant.

It is expected that the odds ratio estimate and its 95% confidence interval are reported for the variable of interest (if it remains in the optimal model) along with these for other variables in the model. The p-values can be reported too.

The authors are encouraged to consult with a statistician in case the regression modelling is of a difficulty.

Response 4: Thanks for your comment. We performed logistic regression model to analyze the role of AR in extrathyroidal extension after consulting Biostatistics Center, Kaohsiung Chang Gung Memorial Hospital. Variables in Table 2 considered for logistic regression models with forward stepwise procedure to analyze if AR is an independent risk factor for extrathyroidal extension. The results of statistical analysis (odds ratio and 95% confidence interval, respectively) showed that independent risk factors for extrathyroidal extension were age (>55 y/o) (OR: 2.88, 95% CI: 1.15-7.2, P = 0.024); lymph node metastasis (OR: 8.56, 95% CI: 3.69-19.94, P < 0.001) and expression of AR (OR: 0.38 (0.17-0.89), 95% CI: 0.17-0.89, P = 0.026). We had modified phrases in result (P5, L21-25) and methodology part (P15, L7-9) and added this result (Table 3) as listed below.

Table 3.  Multiple logistic regression analysis of factors associated with extrathyroidal extension

Variable                

OR (95% CI)

P

Age at diagnosisa

2.88 (1.15- 7.2)

0.024

Sexb

0.52 (0.18-1.47)

0.22

Tumor size c

0.86 (0.31-2.32)

0.762

Lymph node metastasis

8.56 (3.69-19.94)

<0.001

AR expressiond

0.38 (0.17-0.89)

0.026

  1. ≥55 y/o vs. <55 y/o
  2. Male vs. female
  3. >3 cm vs. <3 cm
  4. defined by expression level below/above the median level; high vs. low.

  1. In Supplementary Materials and Methods (p. 1, l. 5), please correct “GSE3678 dataset. We analyzed..” to “GSE3678 dataset, we analyzed…”.

Response 5: Thanks for your comment. We had corrected this grammar error in Supplementary Materials and Methods (P 1, L5).
